# Valorization of Wheat Bran by Three Fungi Solid-State Fermentation: Physicochemical Properties, Antioxidant Activity and Flavor Characteristics

**DOI:** 10.3390/foods11121722

**Published:** 2022-06-13

**Authors:** Ningjie Li, Songjun Wang, Tianli Wang, Rui Liu, Zijian Zhi, Tao Wu, Wenjie Sui, Min Zhang

**Affiliations:** 1State Key Laboratory of Food Nutrition and Safety, Tianjin University of Science and Technology, Tianjin 300457, China; officelnj15@163.com (N.L.); wsj1019111@163.com (S.W.); 13652104287@163.com (T.W.); wutao@tust.edu.cn (T.W.); wjsui@tust.edu.cn (W.S.); 2Food Structure and Function (FSF) Research Group, Department of Food Technology, Safety and Health, Faculty of Bioscience Engineering, Ghent University, Coupure Links 653, 9000 Gent, Belgium; zijian.zhi@ugent.be; 3China-Russia Agricultural Processing Joint Laboratory, Tianjin Agricultural University, Tianjin 300384, China

**Keywords:** wheat bran, solid-state fermentation, Isaria cicadae Miq., Cordyceps militaris, Inonotus obliquus

## Abstract

Three medicinal fungi were used to carry out solid-state fermentation (SSF) of wheat bran. The results showed that the use of these fungi for SSF significantly improved wheat bran’s nutritional properties including the extraction yield of soluble dietary fiber (SDF), total phenolic content (TPC), total flavonoid content (TFC), physical properties containing swelling capacity (SC) and oil absorption capacity (OAC), as well as antioxidant activities. Electronic nose and GC–MS analyses showed that fermented wheat bran had different volatiles profiles compared to unfermented wheat bran. The results suggest that SSF by medicinal fungi is a promising way for the high-value utilization of wheat bran.

## 1. Introduction

Wheat bran is one type of bulk by-product in wheat flour milling and processing. Currently, China is the world’s largest wheat producer and consumer, with an annual wheat output higher than 120 million tons. This results in the annual production of 20–30 million tons of wheat bran in China [1]. Wheat bran is an ideal source of dietary fiber (DF, 35–60%), which is made up of soluble dietary fiber (SDF) and insoluble dietary fiber (IDF). DF is regarded as an important food ingredient for health promotion. SDF is considered to perform more important physiological functions because of its hydrophilic features and availability of probiotics in the intestine [2]. Dietary supplementation of SDF helps maintain blood sugar stability [3]; lowers blood cholesterol levels [4]; regulates intestinal flora and reduces the risk of colorectal cancer [5]. Furthermore, foods rich in SDF enhance satiety and stimulate the release of appetite suppressor hormones, which may be the reason SDF-enriched foods aid in weight loss [6,7].

Many modification technologies have been applied in order to enhance the release of SDF from wheat bran and improve processing adaptability or dietary function. These include either acidic or alkali extraction, enzymatic treatment, fermentation or multiple physical treatments, etc. [6]. Solid-state fermentation (SSF) has gradually attracted the attention of researchers and industry due to its high efficiency, water-saving ability, and low cost [8,9]. At present, solid-state fermentation technology has been industrialized, and is used to convert crops and their waste residue into high-value products such as biosurfactants, pigments, biofertilizers, and organic acids [10,11]. Yeast and lactic acid bacteria (LAB) are frequently used to increase SDF level, phenolics, and nutritional components in wheat bran, as well as bring special wheat flavors, which can be attributed to the aleurone cell walls and proteins of wheat brans were partly degraded after solid-state fermentation [12,13]. Edible fungi other than yeast and LAB have also been used in the SSF of wheat bran. Wheat bran fermented with *Rhizopus oryzae* with 50% water content increased SDF extraction yield from 1.89% to 8.5% [6]. Fermentation with filamentous fungus *Aspergillus niger* was capable of releasing bound phenolic acids from wheat bran, exhibiting good anti-inflammatory activity and better antioxidant ability than standard ferulic acid [14]. Recently, fungi are increasingly used in the fermentation of wheat bran, in order to achieve functional ingredients and healthy foods. For example, the content of total phenol and alkylresorcinols in wheat bran was improved by fermentation with *Fomitopsis pinicola*, and the fermented wheat bran was added to develop whole grain bread with higher nutritional value [15]. The exploration of fungi resources will be beneficial to increasing SDF levels and other nutritional components in wheat bran during SSF.

*Isaria cicadae Miq.* (IC) is an important traditional Chinese medicinal fungus belonging to Cordyceps, which has a history of use of more than 1500 years. IC has been widely studied, third only to *Cordyceps militaris* (CM) and *Ophiocordyceps sinensis,* as they are considered to have special medicinal efficacy and high nutritional value, for instance regulating immune function [16,17,18]. *Inonotus obliquus* (IO) is a white-rot fungus belonging to the *Hymenochaetaceae* Donk, which parasitizes on birches in the cold latitudes of Europe and Asia [19]. IO has been extensively used as a folk medicine in treating tumors, diabetes, and cardiovascular disease [20,21]. Colony photos of three medicinal fungi are shown in Appendix A.

The main purpose of this study was to improve the potential utilization of wheat bran by SSF with three medicinal fungi including *Isaria cicadae Miq.*, *Cordyceps militaris,* and *Inonotus obliquus*. The nutritional, physical, microstructural, antioxidant, and flavor properties of wheat brans were determined after SSF processing.

## 2. Materials and Methods

### 2.1. Materials

The wheat bran was kindly provided by Fada flour company (Dezhou, Shandong Province, China). The wheat bran was finely ground to pass through a 0.45 mm sieve for further use. The chemical composition (%, *w*/*w*) of the wheat bran was 18.31 starch, 14.77 protein, 2.96 fat, 12.28 moisture, 5.74 ash, 42.19 IDF, and 3.75 SDF on a wet weight basis (Appendix A). *Isaria cicadae Miq.* (IC, BNCC 116724), *Cordyceps militaris* (CM, BNCC 190531) and *Inonotus obliquus* (IO, BNCC 117822) were purchased from Beijing Beina Chuanglian Biotechnology Research Institute (Beijing, China). All standard chemicals and other chemical reagents were of analytical grade and were purchased from Sinopharm Chemical Reagent Co., Ltd., company (Tianjin, China).

### 2.2. Solid-State Fermentation (SSF)

The dried wheat bran was mixed with distilled water in the ratio of 1:1 before sterilization at 120 °C for 15 min. The fermented wheat bran was prepared by inoculating 5 g of sterilized wheat bran with pre-cultured IC, CM, and IO spores at 10^7^ cfu/mL (1 mL/10 g dried wheat bran) and then incubating at 27 °C for 8 days. The unfermented sterilized raw wheat bran was designated US-WB and the sterilized wheat bran fermented by IC, CM, and IO strains were designated IC-WB, CM-WB, and IO-WB. The unfermented and fermented wheat bran samples were dried at 45 °C for 48 h, passed through a 0.45 mm sieve, and stored at –20 °C until further analysis.

### 2.3. Determination of SDF

#### 2.3.1. SDF Extraction Yield

SDF from US-WB, IC-WB, CM-WB, and IO-WB was isolated by water extraction and alcohol precipitation method, as described in [22] with some modifications. Briefly, 1 g of wheat bran sample was mixed with 20 mL of distilled water and incubated in a water bath at 85 °C for 2 h under constant stirring. Then, the suspensions were centrifuged at 4000 rpm for 10 min, and the supernatant was precipitated with 4-fold volume of 95% (*v*/*v*) ethanol solution at 4 °C overnight. After centrifugation at 4000 rpm for 20 min, the precipitate was dissolved in water and freeze-dried to obtain SDF. The extraction yield of SDF was expressed as the mass ratio of SDF to the wheat bran sample.

#### 2.3.2. Molecular Weight Distribution

Distribution of molecular weight of SDF was analyzed by a LC-20AT HPLC system (Shimadzu Scientific Instruments, Kyoto, Japan) coupling with RID-10A refractive index detector and OHpak SB-805 HQ column (8.0 × 300 mm, Showa Denko Scientific Instrument Co., Kyoto, Japan), calibrated with T-series dextran standards (Sigma-Aldrich, St Louis, MO, USA). The eluent was ultrapure water and the flow rate was maintained at 0.8 mL/min. The column temperature was 30 °C. SDF samples (1 mg each) were dissolved in 1 mL of ultrapure water and then passed through 0.22-mm membrane filters. The sample injection volume was 20 μL.

## 2.4. Determination of Nutritional Compounds

### 2.4.1. Total Flavonoid Content (TFC)

The determination of TFC was performed according to the method mentioned in [23] with slight modifications. Briefly, 1 g of wheat bran sample was added to 20 mL of 70% (*v*/*v*) methanol solution, ultrasonically extracted for 15 min, and centrifuged at 4000 rpm for 10 min. This procedure was repeated 2 times. Then, the supernatant was mixed with 0.3 mL of sodium nitrite solution (5%, *w*/*w*), 0.3 mL of aluminum nitrate solution (5%, *w*/*w*) and 4.4 mL of sodium hydroxide solution (4%, *w*/*w*). The mixture was incubated at 25 °C for 12 min and then the absorbance was measured at a wavelength of 506 nm. TPC was expressed as rutin equivalent per gram of dried wheat bran sample.

### 2.4.2. Total Phenolic Content (TPC)

TPC was analyzed according to the method described in [24] with slight modifications. Briefly, 1 g of each wheat bran sample was extracted twice with 20 mL of 70% (*v*/*v*) methanol solution for 15 min. The extracts were centrifuged at 4000 rpm for 10 min. Then, 1 mL of the supernatant was mixed with 5 mL of 10% (*w*/*w*) Folin–Ciocalteu reagent and 4 mL of 7.5% (*w*/*w*) sodium carbonate solution. The mixture was incubated in dark at 25 °C for 1 h and then the absorbance was measured at a wavelength of 765 nm. TPC was expressed as gallic acid equivalent per gram of dried wheat bran sample.

### 2.4.3. Alkylresorcinols Content (ARC)

ARC was determined according to the method described by a previous study [25]. The extraction procedure of alkylresorcinols (ARs) was slightly modified, as described in [12]. Briefly, 1 g of each wheat bran sample was extracted with 10 mL acetone at 25 °C for 2 h under constant stirring. The extracts were filtered through a 0.45-µm membrane filter and evaporated to dryness under vacuum at 40 °C. The residue was dissolved in 1 mL of methanol. A 0.4-mL aliquot was added to 4 mL of Fast Blue RR Salt reagent (0.05% Fast Blue RR Salt in methanol), and 20 µL of potassium carbonate (10%, *w*/*v*) as alkalizing agent was added. The mixture was kept in dark at room temperature for 1 h and then the absorbance was measured at a wavelength of 480 nm. ARC was expressed as olivetol equivalent per gram of dried wheat bran sample.

#### 2.4.4. pH Value

The wheat bran sample (10 g) was mixed with 90 mL of distilled water at 25 °C with constant stirring for 30 min. Then, pH was determined by a PHSJ-3F PH meter (INESA INSTRUMENT, Shanghai, China).

### 2.5. Determination of Physical Properties

#### 2.5.1. Water Holding Capacity (WHC)

WHC was analyzed according to the method of [26] with some modifications. The dried wheat bran sample (*m*_1_ = 0.5 g) was soaked in 5 mL of distilled water and shaken for 1 h at 25 °C. After centrifugation at 3000 rpm for 10 min, the supernatant was removed, and then the wet residue was weighed (*m*_2_). WHC was calculated using the following equation:$$\mathrm{WHC}\;(g/g)=\frac{m_{2}-m_{1}}{m_{1}\;}$$

#### 2.5.2. Swelling Capacity (SC)

SC was determined by the method reported by [22] with some modifications. Briefly, 0.5 g of each wheat bran sample (*m*_1_) was weighed in 10-mL glass measuring cylinder and was suspended in 5 mL of distilled water. The suspension was shaken well and stood for 24 h to allow the sample to settle. The volumes of the sample before and after hydration were recorded as *V*_1_ and *V*_2_, and SC was expressed as the following equation:$$\mathrm{SC}\;(\mathrm{mL}/g)=\frac{V_{2}\;-\;V_{1}}{m_{1}}$$

#### 2.5.3. Oil Absorption Capacity (OAC)

OAC was determined according to the method of [27] with a slight modification. The dried wheat bran sample (*m*_1_ = 0.5 g) was added to 5 mL of peanut oil in a 10-mL centrifuge tube. After centrifugation at 3000 rpm for 10 min, the supernatant was removed, and then the wet residue was weighed (*m*_2_). OAC was calculated using the following equation:$$\mathrm{OAC}\;(g/g)=\frac{m_{2}-m_{1}}{m_{1}\;}$$

### 2.6. Microstructure

A JSM-IT300LV scanning electron microscope (SEM) system (JEOL, Japan) was used to obtain SEM images of US-WB and wheat bran samples (IC-WB, CM-WB, and IO-WB) that fermented at 27 °C for 6 days (1000× magnification). Samples were fixed and coated with a thin layer of gold using a JEC-3000FC Auto Fine Coater (JEOL, Japan).

### 2.7. In Vitro Antioxidant Activity Assays

#### 2.7.1. Total Antioxidant Capacity (T-AOC)

The T-AOC test kit was used to determine the antioxidant capacity [15]. Each sample (0.1 g) was added to 1 mL of methanol solution, and the suspension was sonicated at a constant frequency of 40 kHz for 20 min. All steps were performed according to the manufacturer’s instructions. The absorbance values of the extracts were measured at 593 nm, and T-AOC was expressed as Fe^2+^ μmol per gram of dried wheat bran sample.

#### 2.7.2. DPPH Radical Scavenging Assay

The wheat bran sample solution was prepared by following the method stated in [28], slightly modified. Briefly, 1 g of sample powder was ultrasonically extracted twice with 10 mL of 70% (*v*/*v*) methanol at 45 °C for 20 min. The extracts were centrifuged at 4000 rpm for 10 min and the supernatants were collected to obtain the sample solution for further analyses of radical scavenging activities.

The DPPH radical scavenging ability (RSA) was determined by following the method of [29] with slight modifications. Briefly, 0.1 mL of each sample solution was mixed with 3.9 mL of 0.1 mM DPPH methanol solution and allowed to react in the dark for 30 min at room temperature. Subsequently, the absorbance of the test sample (*A*_1_) was measured at 517 nm, and the DPPH solution with nothing added was used as the control sample (*A*_0_). The DPPH RSA of wheat bran extracts was calculated using the following equation:$$\mathrm{DPPH}\;\mathrm{RSA}\;\left(\%\right)=\left(\frac{A_{0}-A_{1}}{A_{0}}\right)\times 100$$

#### 2.7.3. Hydroxyl Radical Scavenging Assay

The hydroxyl (OH) RSA was determined by following the method mentioned in [30] with slight modifications. Briefly, 1 mL of each sample solution was mixed with 1 mL of 9 mM FeSO_4_, 1 mL of 9 mM salicylic acid, and 1 mL of 8.8 mM H_2_O_2_ in the test tube. The mixture was allowed to react at 37 °C for 30 min and the absorbance of the test sample (*A*_1_) was measured at 520 nm. *A*_0_ is the absorbance of the blank (70% (*v*/*v*) methanol instead of sample), *A*_1_ is the absorbance of sample, and *A*_2_ is the background absorbance (70% (*v*/*v*) methanol instead of H_2_O_2_). The OH RSA was calculated using the following equation:$$\mathrm{OH}\;\mathrm{RSA}\;(\%)\;=\;(\frac{A_{2}-A_{1}}{A_{0}})\;\times \;100$$

#### 2.7.4. ABTS Radical Scavenging Assay

The ABTS RSA was determined following the method of [31] with slight modifications. The ABTS working solution was generated by oxidation of 5 mL of 7 mM ABTS with 5 mL of 2.45 mM potassium persulfate in the dark for 16 h and dilution to the absorbance value range of 0.7–0.8 at a wavelength of 734 nm. Each sample (0.4 mL) was reacted with 4 mL ABTS working solution for 6 h at 25 °C in the dark. Subsequently, the absorbance of the test sample (*A*_1_) was measured at 734 nm, and the ATBS working solution was used as the control sample (*A*_0_). The ABTS RSA was calculated using the following equation:$$\mathrm{ABTS}\;\mathrm{RSA}\;(\%)\;=\;(\frac{A_{0}-A_{1}}{A_{0}})\;\times \;100$$

### 2.8. Flavor

#### 2.8.1. Electronic Nose Analysis

An electronic nose system (PEN3, AIRSENSE Analytics, Germany) was used to detect volatile compounds in wheat brans, according to the method described in [32] with slight modifications. The system is equipped with ten metal oxide sensors: W1C, W5S, W3C, W6S, W5C, W1S, W1W, W2S, W2W, and W3S, which are sensitive to aromatic organic compounds, nitrogen oxides, ammonia and aromatic compounds, hydrogen gas, alkanes and aromatic compounds, methane, inorganic sulfur compounds, ethanol, aromatic and organic sulfur compounds, and alkanes, respectively [15]. Volatile compounds were collected by SPME at 60 °C for 30 min. Following injection of each sample onto the electronic nose, data were acquired for 180 s. The carrier gas was maintained at a flow rate of 150 mL/min. Principal component analysis (PCA) and linear discriminant analysis (LDA) were performed to present the volatile flavor difference between wheat bran samples.

#### 2.8.2. Gas Chromatography–Mass Spectrometry (GC–MS) Analysis

The flavor compounds were gathered by solid-phase microextraction (SPME) and analyzed by GC-MS (GCMS-QP2010 Ultral, Shimadzu, Japan) equipped with an Rtx-5MS column (30 m × 0.25 mm × 0.25 µm). The helium was used as the carrier gas with a flow rate of 1.0 mL/min. The injector temperature was 250 °C and the sampling split was 10:1. The column temperature was programmed from 40 °C to 150 °C at 4 °C/min, holding for 1 min at 150 °C, increasing to 250 °C at 8 °C/min with a 6-min hold at 250 °C. The MS spectra were recorded at electron energy of 70 eV, and ion source temperature was 200 °C, scanned in the *m*/*z* 35–500. Volatile compounds with mass spectral match factors over 85 in NIST14 library were identified. The volatile components in wheat bran samples and their relative contents were determined by peak area normalization.

### 2.9. Statistical Analysis

Unless otherwise specified, three independent trials were performed, using a new batch for each sample preparation. Data were expressed as mean ± standard deviation, and the results were analyzed using Tukey’s test with a significance level of *p* < 0.05. Principal component analysis (PCA) and linear discriminant analysis (LDA) of electronic nose data were performed using Winmuster (Airsense Analytics GmbH, Schwerin, Germany) version 1.6.2.18.

## 3. Results and Discussion

### 3.1. Extraction Yields and Molecular Weight Distributions of Soluble Dietary Fibers

The SDF extraction yield of IC-WB, CM-WB, and IO-WB samples is shown in Figure 1A. The extraction yield of SDF increased significantly from 5.60% to 11.02% (US-WB) after autoclaving, indicating that hydrothermal treatment could enhance the cleavage of the cell wall and release more SDF [33]. The SDF extraction yield of IC-WB, CM-WB, and IO-WB increased to 13.39%, 13.18%, and 13.67%, respectively, after 6 days of fermentation. Afterward, the SDF extraction yield of all fermented wheat bran showed a downward trend, possibly due to the substrate that was consumed, which could not provide the nutrients needed for mycelial growth, and microorganisms began to use SDF. An increase in SDF level was found in wheat bran fermented with either *Rhizopus oryzae*, yeast, or lactic acid bacteria [6,12]. The increase in SDF release may be explained as the degradation of cellulose and hemicellulose and the formation of loose structures during the SSF process, thereby causing more soluble polysaccharides to be released [34]. While the decrease in SDF extraction yield after 6 days of fermentation suggests the degradation of soluble polysaccharides carried out by the enzymatic activities of the microorganisms during SSF [27].

The molecular weight distributions of unfermented and fermented wheat brans with 6 days of fermentation are shown in Figure 1B and Appendix A. The molecular weight of US-WB is mainly concentrated above 2 × 10^5^ Da, and the *M_w_* values and its related percentage are, respectively, >2 × 10^6^ Da (38.43%), 2 × 10^5^–2 × 10^6^ Da (37.08%), 1 × 10^4^–2 × 10^5^ Da (17.73%) and <1 × 10^4^ Da (6.76%). The molecular weight of IC-WB is mainly concentrated above 2 × 10^6^ Da, and the *M_w_* values and its related percentages are, respectively, >2 × 10^6^ Da (82.53%), 2 × 10^5^–2 × 10^6^ Da (13.95%), and 1 × 10^4^–2 × 10^5^ Da (3.52%). Compared with US-WB, the relative peak area percentages (*M*_w_ < 1×10^4^ Da) of CM-WB and IO-WB significantly increased to 12.75% and 20.30%, respectively, while the peak area percentages (*M*_w_ = 1×10^4^–2 × 10^5^ Da) decreased correspondingly (Appendix A). Notably, the molecular weights of SDF typically varied from 1 × 10^4^ Da to 7.2 × 10^6^ Da for extractable AXs in previous reports [35]. In the present work, the fermented wheat bran clearly displayed a higher degree of hydrolysis, compared with US-WB. Additionally, applying different microorganisms led to obviously different fermentation characteristics, which was in agreement with the results of SDF extraction yield. SDF fractions from IC-WB had a relatively narrow *M*_w_ distribution, suggesting that AXs and β-glucans were effectively degraded or depolymerized upon fermentation.

### 3.2. Nutritional Properties of Wheat Brans

Regarding TFC results, it was observed that TFC reached maximum values of 3.18 mg/g (IC-WB), 1.97 mg/g (CM-WB), and 2.41 mg/g (IO-WB) after 6 days of fermentation (Figure 2A). According to the report by [36], fermentation could promote the release of flavonoids and other active components in wheat bran as a result of the increase in the specific surface area of wheat bran and the acid hydrolysis that occurs during fermentation. Similarly, TPC increased after fermentation in the order as follows: IC-WB > CM-WB > IO-WB. The highest TPC values were observed in IC-WB (3.18 mg/g), CM-WB (2.18 mg/g), and IO-WB (2.12 mg/g) after 6 days of fermentation. It has been reported that SSF can promote the conversion of bound phenolics into free phenolics, thus increasing their release and bioavailability [37]. According to [38], the release of phenolic acids from wheat bran could be improved under acidic hydrolysis conditions. The pH of IC-WB was lower than that of CM-WB and IO-WB, which might result in the enhanced release of phenolic acids via the acidic hydrolysis of wheat bran.

ARs are a group of phenolic lipids mainly found in the outer layer of wheat and rye kernels and are considered to be biomarkers for the intake of whole grain foods [39]. As shown in Figure 2C, the ARC of all fermented wheat brans first increased and then decreased with the extension of fermentation time. Among them, IC-WB exhibited the highest ARC of 0.68 mg/g at 4 days of fermentation, while CM-WB and IO-WB reached the maximum value of 0.69 mg/g and 0.77 mg/g at 6 days of fermentation. Correspondingly, the pH value of IC-WB was reduced after 4 days of fermentation (Figure 2D). This result is consistent with a report that pH has an influence on AR content, and ARs were severely degraded at low pH [24].

### 3.3. Functional Properties

Water holding capacity (WHC), swelling capacity (SC), and oil absorption capacity (OAC) of fermented wheat brans are shown in Table 1. Except for WHC of *Isaria cicadae Miq*. fermented wheat bran (IC-WB), WHC, SC and OAC values of all fermented wheat brans generally increased with time. This result can be attributed to the collapse of firm cell wall structures and the increase in SDF transferred from IDF because SDF can absorb more water than IDF [15]. IC-WB had no significant changes in WHC upon 6 days of fermentation and an obvious decrease in WHC at the 8 days of fermentation. This decrease in WHC could be due to the degradation of soluble polysaccharides, which are macromolecular sugar chains with many hydroxyl groups and exhibit strong water-holding capacity [40]. Additionally, WHC and SC of IC-WB were significantly lower than that of *Cordyceps militaris* fermented wheat bran (CM-WB) and *Inonotus obliquus* fermented wheat bran (IO-WB), especially on the 8th day of fermentation. In contrast, IC-WB had significantly higher OAC compared to other fermented wheat brans. The highest OAC was observed in IC-WB fermented for 8 days with 3.28 g/g. The increase in OAC could be owing to the loose structure generated during fermentation, as shown in Figure 3, and the existing SDF may be another reason for the increase in OAC because some components of dietary fiber have a strong affinity for lipid material [34], which will bring benefits to the processing of high-fat foods.

### 3.4. Morphology

The morphologies of the SDF of fermented wheat brans were examined by SEM (Figure 3), which revealed that starch granules and adherent endosperm protein were mostly removed from the aleurone layer of fermented wheat brans compared to US-WB [41]. Notably, some filamentous structures were observed in IC-WB and the fibrous cellulose was exposed after microbial fermentation. For CM-WB, rod-like structures appeared with an uneven surface, whereas IO-WB showed a flat surface with large amounts of short rods. These differences in microstructural features of fermented wheat brans may be reflected in different components of enzymes secreted by IC, CM, and IO [42]. Thus, further studies on enzyme activities of amylase, protease, cellulase, etc., are needed to facilitate research on the fermentation process of wheat bran. In any case and to different degrees, fermentation can destroy the rigid structures of wheat bran tissue, thereby releasing more nutrients [43,44]. These results were consistent with the above analysis of the nutritional properties of wheat brans.

### 3.5. In Vitro Antioxidant Activities

Due to the structural destruction of wheat brans during fermentation, a large number of antioxidant compounds such as phenolics, flavonoids, etc., were released from wheat bran cell walls [45], thereby enhancing the antioxidant abilities of wheat brans. As shown in Figure 4A, T-AOC of fermented wheat brans increased with time and reached the highest values of 34.61 µmol/g (IC-WB), 21.03 µmol/g (CM-WB), and 21.59 µmol/g (IO-WB) at day 6 of fermentation. Compared to CM-WB and IO-WB, the T-AOC of IC-WB was significantly enhanced and represented two times as much as that of sterilized wheat bran (17.19 µmol/g).

The antioxidant activities of fermented wheat bran can be characterized to a certain extent by DPPH RSA, OH RSA, and ABTS RSA [46,47]. In Figure 4B–D, all radical scavenging activities of fermented wheat brans were enhanced with the increase in the fermentation time before day 6 of fermentation, and then RSA of all radicals was reduced on day 8 of fermentation. This trend was in agreement with that of TFC and TPC [48]. In addition to T-AOC, all radicals RSA increased after fermentation in the order as follows: IC-WB > CM-WB > IO-WB. The highest radicals RSA was observed in IC-WB that fermented for 6 days with DPPH RSA of 21.17%, OH RSA of 28.54%, and ABTS RSA of 75.19%, which was 2.05 times, 2.57 times, and 2.09 times as much as that of sterilized wheat bran, respectively. This result could be attributed to the fact that more phenolics and flavonoids were released from IC-WB after fermentation in comparison with CM-WB and IO-WB.

### 3.6. Flavor

#### 3.6.1. Electronic Nose Analysis

Principal component analysis (PCA) and linear discriminant analysis (LDA) were applied to convert high-dimensional data from different sensors to two-dimensional data for convenience in data analysis. In Figure 5, PCA showed an accumulated contribution of 92.19% with the first and second principal components (PC1 and PC2) being 79.43% and 12.76%, while LDA had an accumulated contribution of 97.71% with the contribution from LDA1 and LDA2 of 94.96% and 2.75%, respectively. In general, the larger the distance between two clusters in the LDA graph, the greater the flavor difference between the samples. The result indicated that wheat bran fermented with the respective fungi had different volatile flavor characteristics.

#### 3.6.2. GC–MS Analysis

Table 2 lists the data of volatile compounds in US-WB and fermented wheat brans upon fermentation for 6 days. A total of 56 volatile compounds were detected, including 15 alcohols, 4 ketones, 5 acids, 5 esters, 7 aldehydes, 16 hydrocarbons, 1 phenol, and 3 other compounds. Among them, 40, 23, 33, and 30 volatile compounds were identified in US-WB, IC-WB, CM-WB, and IO-WB, respectively, suggesting that fermentation could change the compositions of volatile compounds in different wheat brans. Kinds of alcohols, acids, and aldehydes decreased while ketones and esters increased after fermentation. Ethanol disappeared in IC-WB and IO-WB, which might be attributed to the fact that alcohols reacted with acids to form esters [12]. Hexanol with a fruity aroma was detected in all wheat bran samples, whereas isoamyl butyrate, associated with banana and pear aromas, increased after fermentation. After fermentation, 7-hydroxy-3,7-dimethyl-octanal, the aroma of lily and bell orchid, appeared in the fermented rather than unfermented wheat bran. The 1-Hexanol flavor was enhanced after fermentation; it has a special flavor and is mainly used to prepare coconut and berry essence. Therefore, the formation of some aroma components in fermented wheat brans is conducive to food flavoring.

## 4. Conclusions

All three medicinal fungi *Isaria cicadae Miq.*, *Cordyceps militaris,* and *Inonotus obliquus* improved the level of soluble dietary fiber (SDF), total flavonoid content (TPC), total phenolic content (TFC), swelling capacity (SC), oil absorption capacity (OAC) and antioxidant activities of wheat brans during SSF processing. Notably, *Isaria cicadae Miq.* exhibited different fermentation characteristics than the others, achieving an SDF fraction with an average *M*_w_ of 3.19 × 10^6^ Da. In addition, wheat bran fermented with the respective fungi had different volatile profiles. The results indicate that the exploration of more fungi resources was an efficient strategy for solid-state fermentation to valorize wheat bran and to develop functional whole grain foods.

## Figures and Tables

**Figure 1 foods-11-01722-f001:**
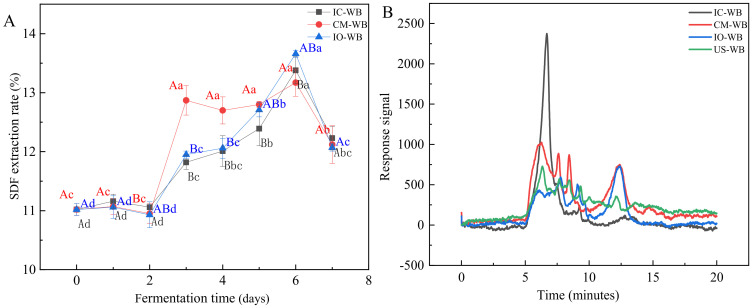
The extraction yield and molecular weight of soluble dietary fiber (SDF) in fermented wheat brans made using different microorganisms. (**A**) The extraction yield, (**B**) the molecular weight. Different lowercase letters indicate significant differences at *p* < 0.05 in terms of fermentation time. Different capital letters indicate significant differences at *p* < 0.05 in terms of microorganisms. IC-WB, *Isaria cicadae Miq.* fermented wheat bran; CM-WB, *Cordyceps militaris* fermented wheat bran; IO-WB, *Inonotus obliquus* fermented wheat bran; US-WB, unfermented sterilized wheat bran.

**Figure 2 foods-11-01722-f002:**
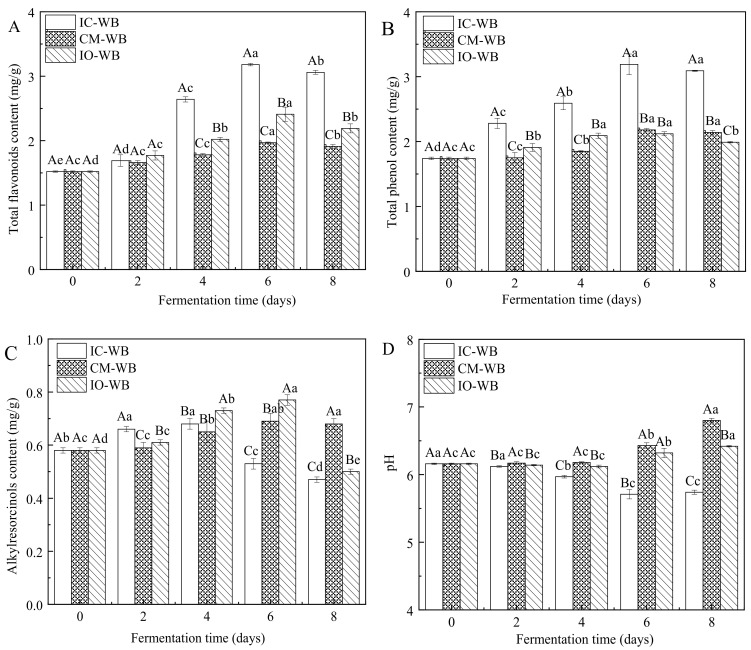
Changes of nutritional compounds in fermented wheat brans. (**A**) Total flavonoid content (TFC), (**B**) total phenolic content (TPC), (**C**) alkylresorcinols content (ARC) and (**D**) pH value. Different lowercase letters indicate significant differences at *p* < 0.05 in terms of fermentation time. Different capital letters indicate significant differences at *p* < 0.05 in terms of microorganisms. IC-WB, *Isaria cicadae Miq.* fermented wheat bran; CM-WB, *Cordyceps militaris* fermented wheat bran; IO-WB, *Inonotus obliquus* fermented wheat bran.

**Figure 3 foods-11-01722-f003:**
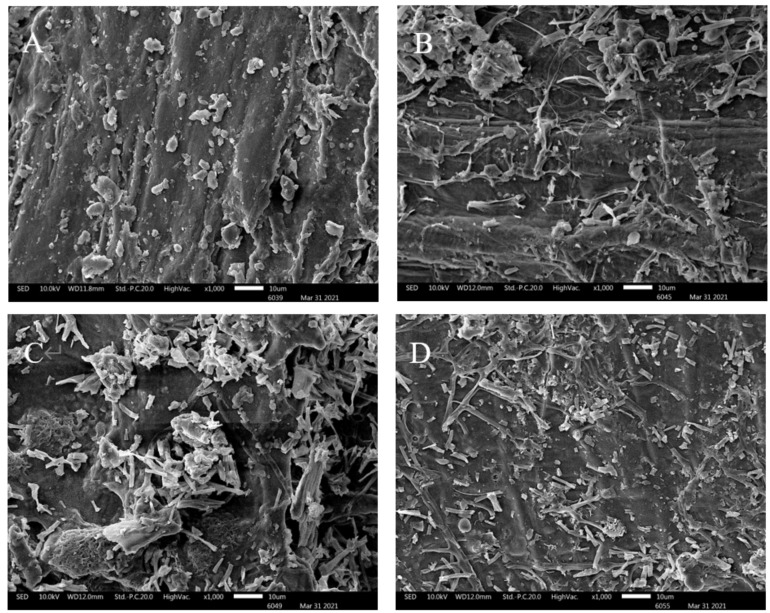
Scanning electron microscope (SEM) images of the US-WB (**A**), IC-WB (**B**), CM-WB (**C**), and IO-WB (**D**) after 6 days of fermentation. US-WB, unfermented sterilized wheat bran. IC-WB, *Isaria cicadae Miq.* fermented wheat bran; CM-WB, *Cordyceps militaris* fermented wheat bran; IO-WB, *Inonotus obliquus* fermented wheat bran.

**Figure 4 foods-11-01722-f004:**
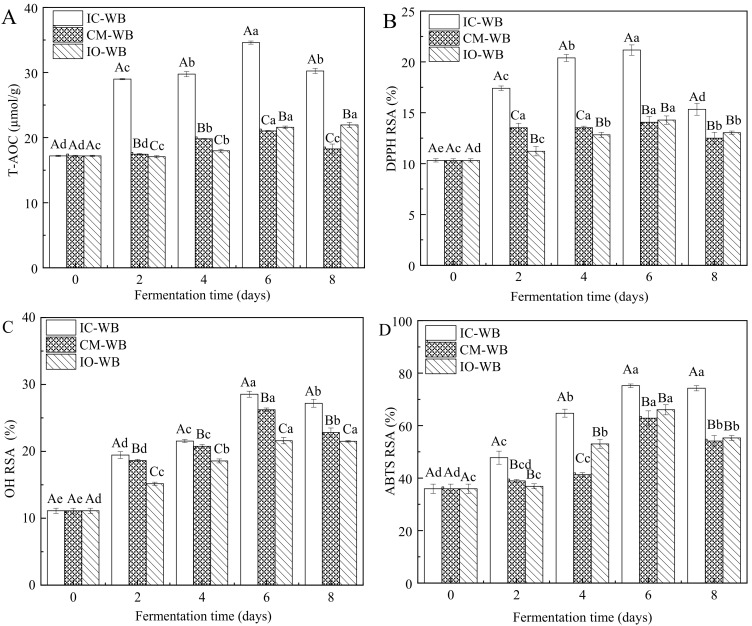
In vitro antioxidant activities of fermented wheat brans characterized by (**A**) total antioxidant capacity (T-AOC), (**B**) DPPH radical scavenging ability (DPPH RSA), (**C**) hydroxyl radical scavenging ability (OH RSA), and (**D**) ABTS radical scavenging ability (ABTS RSA). Different lowercase letters indicate significant differences at *p* < 0.05 in terms of fermentation time. Different capital letters indicate significant differences at *p* < 0.05 in terms of microorganisms. IC-WB, *Isaria cicadae Miq.* fermented wheat bran; CM-WB, *Cordyceps militaris* fermented wheat bran; IO-WB, *Inonotus obliquus* fermented wheat bran.

**Figure 5 foods-11-01722-f005:**
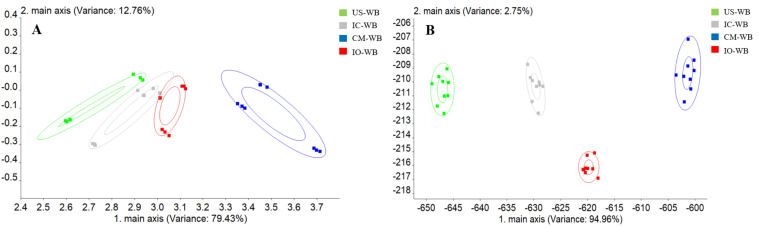
Principal component analysis (**A**) and linear discriminant analysis (**B**) of unfermented and fermented wheat brans after 6 days of fermentation. US-WB, unfermented sterilized wheat bran. IC-WB, *Isaria cicadae Miq.* fermented wheat bran; CM-WB, *Cordyceps militaris* fermented wheat bran; IO-WB, *Inonotus obliquus* fermented wheat bran.

**Table 1 foods-11-01722-t001:** Water holding capacity (WHC), swelling capacity (SC), and oil absorption capacity (OAC) of fermented wheat brans at different fermentation times.^a.^

Samples	WHC (g/g)			SC (mL/g)			OAC (g/g)		
	IC-WB	CM-WB	IO-WB	IC-WB	CM-WB	IO-WB	IC-WB	CM-WB	IO-WB
0 d	3.06 ± 0.07 ^Aa^	3.06 ± 0.07 ^Ab^	3.06 ± 0.07 ^Ad^	2.27 ± 0.05 ^Abc^	2.27 ± 0.05 ^Ad^	2.27 ± 0.05 ^Ad^	2.24 ± 0.03 ^Ad^	2.24 ± 0.03 ^Ac^	2.24 ± 0.03 ^Ad^
2 d	3.15 ± 0.03 ^Ba^	3.47 ± 0.08 ^Aa^	3.54 ± 0.03 ^Aa^	2.25 ± 0.02 ^Bc^	2.51 ± 0.01 ^Ac^	2.52 ± 0.06 ^Ac^	2.27 ± 0.02 ^Ad^	2.36 ± 0.04 ^Ab^	2.27 ± 0.04 ^Ad^
4 d	3.10 ± 0.05 ^Ba^	3.43 ± 0.02 ^Aa^	3.48 ± 0.03 ^Aab^	2.34 ± 0.02 ^Bb^	2.62 ± 0.01 ^Ab^	2.64 ± 0.04 ^Ab^	2.54 ± 0.05 ^Ac^	2.35 ± 0.05 ^Bb^	2.48 ± 0.02 ^Ac^
6 d	3.10 ± 0.06 ^Ba^	3.40 ± 0.04 ^Aa^	3.44 ± 0.02 ^Abc^	2.52 ± 0.07 ^Ba^	2.78 ± 0.05 ^Aa^	2.76 ± 0.04 ^Aa^	3.10 ± 0.04 ^Ab^	2.45 ± 0.01 ^Ca^	2.69 ± 0.01 ^Bb^
8 d	2.52 ± 0.02 ^Bb^	3.39 ± 0.04 ^Aa^	3.38 ± 0.05 ^Ac^	1.43 ± 0.03 ^Cd^	2.81 ± 0.08 ^Aa^	2.56 ± 0.04 ^Bbc^	3.28 ± 0.03 ^Aa^	2.40 ± 0.03 ^Cb^	2.78 ± 0.01 ^Ba^

^a^ Data are expressed as the mean ± SD (n = 3). Different lowercase letters indicate significant differences at *p* < 0.05 in terms of fermentation time. Different capital letters indicate significant differences at *p* < 0.05 in terms of microorganisms. IC-WB, *Isaria cicadae Miq.* fermented wheat bran; CM-WB, *Cordyceps militaris* fermented wheat bran; IO-WB, *Inonotus obliquus* fermented wheat bran.

**Table 2 foods-11-01722-t002:** Volatile compounds in unfermented sterilized wheat bran (US-WB) and fermented wheat brans after 6 days of fermentation. IC-WB, *Isaria cicadae Miq.* fermented wheat bran; CM-WB, *Cordyceps militaris* fermented wheat bran; IO-WB, *Inonotus obliquus* fermented wheat bran.

RT	CAS	Compounds	Relative Content (%)			
			US-WB	IC-WB	CM-WB	IO-WB
Alcohols			34.86	22.01	45.87	34.79
1.524	64-17-5	Ethanol	1.00	—	1.83	—
1.7	75-65-0	2-Propanol, 2-methyl-	2.61	8.35	4.31	4.04
2.871	107-98-2	2-Propanol, 1-methoxy-	3.82	—	0.96	2.20
3.914	763-32-6	3-Buten-1-ol, 3-methyl-	1.32	4.65	2.01	1.75
4.825	71-41-0	1-Pentanol	—	—	2.08	2.80
5.172	107-41-5	Hexylene glycol	1.00	—	—	—
7.631	98-00-0	2-Furanmethanol	1.14	—	—	—
8.153	111-27-3	1-Hexanol	0.68	6.67	5.94	5.51
8.6	19550-89-1	2,2-Dimethyl-5-hexen-3-ol	1.17	—	—	—
8.616	59562-82-2	1,2-Butanediol, 3,3-dimethyl-	—	—	2.37	2.51
11.909	54004-46-5	2H-Pyranmethanol, tetrahydro-2,5-dimethyl-	14.13	—	16.50	15.98
12.514	3391-86-4	1-Octen-3-ol	1.25	—	1.43	—
15.639	2050-95-5	1-Butanol, 3-methyl-, carbonate (2:1)	1.76	—	—	—
21.095	41902-42-5	3-Pentanol, 3-(1,1-dimethylethyl)-2,2,4,4-tetramethyl-	4.98	—	6.97	—
30.12	2425-77-6	1-Decanol, 2-hexyl-	—	2.34	1.47	—
Ketones			1.24	10.69	2.56	2.95
2.029	78-94-4	Methyl vinyl ketone	—	2.54	1.09	1.34
10.394	110-13-4	2,5-Hexanedione	1.24	3.45	1.47	1.61
17.185	1123-09-7	2-Cyclohexen-1-one, 3,5-dimethyl-	—	2.17	—	—
24.788	112-12-9	2-Undecanone	—	2.53	—	—
Acids			30.00	8.95	3.71	3.92
2.028	64-19-7	Acetic acid	10.87	—	—	—
7.168	503-74-2	Butanoic acid, 3-methyl-	0.98	—	—	—
12.609	504-85-8	3-Pentenoic acid, 4-methyl-	3.63	8.95	3.71	3.92
12.882	142-62-1	Hexanoic acid	13.46	—	—	—
23.849	112-05-0	Nonanoic acid	1.06	—	—	—
Esters			0.91	7.53	7.68	9.45
9.362	4435-53-4	1-Butanol, 3-methoxy-, acetate	—	3.86	—	—
12.156	595-37-9	Propanoic acid, 2-methyl-, 3-methylbutyl ester	—	—	5.65	5.53
15.469	695-06-7	2(3H)-Furanone, 5-ethyldihydro-	0.91	—	—	—
15.653	106-27-4	Butanoic acid, 3-methylbutyl ester	—	3.67	2.03	2.31
30.128	6222-02-2	Tetradecyl trifluoroacetate	—	—	—	1.61
Aldehydes			9.45	11.01	13.47	14.61
3.227	110-62-3	Pentanal	0.85	—	1.66	1.85
5.7	66-25-1	Hexanal	3.76	1.47	5.22	5.81
6.812	98-01-1	Furfural	0.84	—	—	—
9.344	111-71-7	Heptanal	0.72	—	—	—
16.1	107-75-5	Octanal, 7-hydroxy-3,7-dimethyl-	—	9.54	5.15	5.38
17.553	124-19-6	Nonanal	2.10	—	1.44	1.57
27.616	13019-16-4	2-Octenal, 2-butyl-	1.18	—	—	—
Hydrocarbons			19.23	36.05	20.84	29.50
7.384	7154-80-5	Heptane, 3,3,5-trimethyl-	—	1.78	0.92	—
7.784	100-41-4	Ethylbenzene	0.67	—	—	0.94
8.079	108-38-3	Benzene, 1,3-dimethyl-	1.20	2.40	1.19	1.38
8.885	629-20-9	1,3,5,7-Cyclooctatetraene	2.10	—	—	—
8.909	100-42-5	Styrene	—	1.93	1.00	0.96
12.93	13475-82-6	Heptane, 2,2,4,6,6-pentamethyl-	—	—	0.98	2.13
13.955	20278-85-7	Heptane, 2,3,5-trimethyl-	—	—	—	1.01
14.303	527-84-4	o-Cymene	1.58	2.79	1.91	2.11
14.48	5989-27-5	D-Limonene	0.70	—	0.97	1.53
20.184	1002-43-3	Undecane, 3-methyl-	1.74	2.88	1.86	3.11
21.311	112-40-3	Dodecane	4.72	9.20	5.39	7.85
26.286	20959-33-5	Heptadecane, 7-methyl-	—	1.57	—	1.01
27.8	3856-25-5	Copaene	1.46	—	—	—
28.485	629-59-4	Tetradecane	2.88	9.66	4.75	5.55
33.961	638-36-8	Hexadecane, 2,6,10,14-tetramethyl-	0.84	3.84	1.87	—
34.701	629-94-7	Heneicosane	1.34	—	—	1.92
Phenols			0.96	—	—	—
25.528	7786-61-0	2-Methoxy-4-vinylphenol	0.96	—	—	—
Other compounds			4.31	3.76	5.87	4.78
13.004	3777-69-3	Furan, 2-pentyl-	1.08	—	—	—
20.626	91-20-3	Naphthalene	1.26	3.76	1.12	—
27.477	13187-99-0	2-Bromo dodecane	1.97	—	4.75	4.78

## Data Availability

No new data were created or analyzed in this study. Data sharing is not applicable to this article.

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
