# Peer review of "Valorization of Wheat Bran by Three Fungi Solid-State Fermentation: Physicochemical Properties, Antioxidant Activity and Flavor Characteristics"

_foods, 2022, doi:10.3390/foods11121722_

Round 1

Reviewer 1 Report

Manuscript Number: foods-1738228

Title: Valorization of wheat bran by three fungi Solid state fermentation :SDF、physico-chemical properties and antioxidant activity

Overview and general recommendation

 The title is specific and reflects the main ideas of the article. The article structure is compact, sequential and logical. The data are adequate to support the conclusion. The methods section provides sufficient information on design, sampling, definitions, data collection and data analysis. References are up-dated adequate and correctly cited, but there are too few references in total.

The article presents the use of SSF for the multiplication of medicinal mushrooms as a promising way to increase the nutritional value of wheat bran.

Minor comments:

  1. The part of Introduction: can be improved with information from the specialty literature about the three medicinal fungi (if possible photos of the three medicinal fungi).
  2. The part of Results and Discussion:
    • 1. Figure 1 (in what time is it expressed? hours, days etc.) to move after text
    • pay close attention to the numbering of the tables: Table S1 appears in the text (line 263), then Table 2 in which WHC, SC and OAC are shown ??? (line 308)
  3. The part of Conclusions:
  • In this part I recommend not to use abbreviations.
  • In my opinion, the information on improving the nutritional value of fermented wheat bran should be detailed.

 Overall, the article was easy to read, to understand, the authors carry out an interesting work.

Author Response

        Many thanks for your insightful and helpful comments and suggestions. We have carefully carried out a diligent revision of the whole paper. The point-by-point answers to these comments and suggestions are listed in the word. Please see the attachment.

Reviewer 2 Report

Dear authors,

Fungi effect after solid state fermentation on wheat bran has been well documented in your paper as you have carried out a deep characterization of the material after the process. 

I would recommend you to give a little bit of discussion comparing with other fermented material in order to know if the fungi you used provided similar results. Also, I would include the complete name in some parts of the paper as, in my opinion, when many abbreviations are used is confusing.

Also, I recommend some modifications:

There is a mistake on the title

Line 29. Eliminate “DF”

Line 40-41. Include some industry uses of SSF

Line 42-43. Explain how SDF is increased by yeasts and lactic acid bacteria

Line 45. Wheat bran in lower case

Line 45. Rhizopus oryzae in italic. In general all microorganisms names in the text in capital letter.

Line 49. Give an example

Line 50. Explain the meaning of medicinal fungi

Line 69-70. It is better to put the composition in a table. The sum of all the components must be 100 %.

Line 70-71. Give the name of the specific strain of each fungus.

Line 76. Indicate the solution used to fit the initial moisture content

Line 76. Include sterilization conditions

Line 77. Include preculture preparation. What did you used?, glycerinated spores?, spores concentration for the inoculation?

Line 79. Why did you designated the sterilized raw wheat bran as US-WB

Line 81-82. Non fermented was not dried?

Line 148. “Alkalizing” in lower case

Line 153. When you say “wheat bran sample” you mean fermente and no-fermented, I suppose

Line 153. I do not understand why you measure pH during constant stirring. Does it change a lot?. Maybe it would be better do the mixing well and then, measure the pH.

Line 166. Glass cylinder?

Line 178. I would write JSM-IT300LV scanning electron microscope (SEM)

Line 208. Change the position of “and”

Line 209. Subscript in H2O2

2.8.1 and 2.8.2. Explain differences in the compounds detected with each technique

Figure 1 leyend. Indicate the three studied fungi

Line 268. “the molecular weight is less than 1×104 Da without distribution”. This is not clear

Line 269-271. Phrase needs revision.

In section 3.1 It is not clear that the particle distribution is considered a nutritional property to be included in the section. Why did you study the particle distribution? Is there a positive effect of particle reduction?

Figure 2 caption. I would not use abbreviations

Line 284. Delete “of all fermented wheat brans” as it is confusing.

Figur-e 2 A and figure 2B. Could you explain maximum values attained with IC?

Line 308-312. In my opinion, too many abbreviations.

Line 306. Different capital letters indicate significant differences… You should specify

Line 313. Could you explain the decrease of WHC the 8 day of fermentation?

Line 323. Lower case in Lipid

Line 331. It is not clear in the image “ a flat surface with some small pores”

Line 352. Where is the value of  sterilized wheat bran.

Line 357. Why the activity is reduced the 8th day?

Table 2. It could be interesting to include aroma descriptors of some volatile compounds, either on the table or in the description.

Author Response

Many thanks for your insightful and helpful comment and suggestions. We have carefully carried out a diligent revision of the whole paper. The point-by-point answers to these comments and suggestions are listed in the word. Please see the attachment.

Reviewer 3 Report

The research is interesting and novel. There are a few aspects that can be improved. There are some information lacking from the materials and methods that should be added. Please see my detailed comments in the attachment. The figure captions and table headings need to be improved and abbreviations in figures and tables should be explained. Table numbering should be corrected and supplementary table cited, but not included. The conclusions can also be improved. Incorrect formatting was applied to the reference section and needs to be corrected.

Author Response

Many thanks for your insightful and help comments and suggestions. We have carefully carried out a diligent revision of the whole paper. The point-by-point answers to these comments and suggestions are listed in the word. Please see the attachment. 

Round 2

Reviewer 3 Report

Thank you for making the necessary changes and improvements to the manuscript. However, there are still a few minor issues that needs to corrected. Please consult the editor about the references. There are inconsistencies with other articles published in the same journal. Attached, please find my detailed comments.

Author Response

(The authors gave the same response as above.)
